

**Spectral- and size-resolved mass absorption efficiency of mineral dust aerosols in**
**the shortwave: a simulation chamber study**
Lorenzo Caponi[1,2], Paola Formenti[1], Dario Massabó[2], Claudia Di Biagio[1], Mathieu Cazaunau[1], Edou-
ard Pangui[1], Servanne Chevaillier[1], Gautier Landrot [3], Meinrat O. Andreae[4,11], Konrad Kandler[5], Stuart
Piketh[6], Thuraya Saeed[7], Dave Seibert[8], Earl Williams[9], Yves Balkanski[10], Paolo Prati[2], and Jean-
François Doussin[1]
*[1] Laboratoire Interuniversitaire des Systèmes Atmosphériques (LISA), UMR 7583, CNRS, Université Paris-Est-
Créteil et Université Paris Diderot, Institut Pierre Simon Laplace, Créteil, France*
*[2] University of Genoa, Department of Physics & INFN, Genoa, Italy*
*[3] Synchrotron SOLEIL, L'Orme des Merisiers Saint-Aubin, France*
*[4] Biogeochemistry Department, Max Planck Institute for Chemistry, P.O. Box 3060, 55020, Mainz, Germany*
*[5] Institut für Angewandte Geowissenschaften, Technische Universität Darmstadt, Schnittspahnstr. 9, 64287
Darmstadt, Germany*
*[6] Climatology Research Group, University of the Witwatersrand, Johannesburg, South Africa*
*[7] Science department, College of Basic Education, Public Authority for Applied Education and Training, Al-
Ardeya, Kuwait*
*[8] Walden University, Minneapolis, Minnesota, USA*
*[9] Massachusetts Institute of Technology, Cambridge, Massachusetts, USA*
*[10] LSCE, CNRS UMR 8212, CEA, Université de Versailles Saint-Quentin, Gif sur Yvette, France*
*[11]Geology and Geophysics Department, King Saud University, Riyadh, Saudi Arabia*
* Corresponding author: paola.formenti@lisa.u-pec.fr





**Abstract**
This paper presents new laboratory measurements of the mass absorption efficiency (MAE) between
375 and 850 nm for mineral dust of different origin in two size classes: $PM_{10.6}$ (mass fraction of particles
of aerodynamic diameter lower than 10.6 μm) and $PM_{2.5}$ (mass fraction of particles of aerodynamic
diameter lower than 2.5 μm). Experiments have been performed in the CESAM simulation chamber
using generated mineral dust from natural parent soils, and optical and gravimetric analyses.
Results show that the MAE values are lower for the $PM_{10.6}$ mass fraction (range 37-135 $10^{-3}$ $m^2\,g^{-1}$ at
375 nm) than for the $PM_{2.5}$ (range 95-711 $10^{-3}$ $m^2\,g^{-1}$ at 375 nm), and **decrease** with increasing wave-
length as $\lambda^{-AAE}$, where Angstrom Absorption Exponent (AAE) averages between 3.3-3.5, regardless of
size. The size-independence of AAE suggests that, for a given size distribution, the possible variation of
dust composition with size would not affect significantly the spectral behavior of shortwave absorption.
Because of its high atmospheric concentration, light-absorption by mineral dust can be competitive to
black and brown carbon even during atmospheric transport over heavy polluted regions, when dust con-
centrations are significantly lower than at emission. The AAE values of mineral dust are higher than for
black carbon (~1), but in the same range as light-absorbing organic (brown) carbon. As a result, depend-
ing on the environment, there can be some ambiguity in apportioning the AAOD based on spectral de-
pendence, which is relevant to the development of remote sensing of light-absorption aerosols from
space, and their assimilation in climate models. We suggest that the sample-to-sample variability in our
dataset of MAE values is related to regional differences of the mineralogical composition of the parent
soils. Particularly in the $PM_{2.5}$ fraction, we found a strong linear correlation between the dust light-
absorption properties and elemental iron rather than the iron oxide fraction, which could ease the appli-
cation and the validation of climate models that now start to include the representation of the dust com-
position, as well as for remote sensing of dust absorption in the UV-VIS spectral region.
**1. Introduction**
Mineral dust aerosols emitted by wind erosion of arid and semi-arid soils account for about 40% of the
total emitted aerosol mass per year at the global scale (Knippertz and Stuut, 2014). The episodic but
frequent transport of intense mineral dust plumes is visible from spaceborne sensors as their high con-
centrations, combined to their ability of scattering and absorbing solar and thermal radiation, give raise
to the highest registered values of aerosol optical depth (AOD) on Earth (Chiapello, 2014). The instan-
taneous radiative efficiency of dust particles, that is, their radiative effect per unit AOD, is of the order





of tenths to hundreds of W m$^{-2}$ AOD$^{-1}$ in the solar spectrum, and of the order of order of tenths W m$^{-2}$
AOD$^{-1}$ in the thermal infrared (e.g., Haywood et al., 2003; di Sarra et al., 2011; Slingo et al., 2006 and
the compilation of Highwood and Ryder, 2014). In the solar spectrum, (Boucher et al., 2013). Albeit
partially compensated by the radiative effect in the thermal infrared, the global mean radiative effect of
mineral dust in the shortwave is negative both at the surface and the top of the atmosphere (TOA) and
local warming of the atmosphere. Many are the consequences on the global and regional climate, that
ultimately feed back on wind speed and vegetation and therefore on dust emission (Tegen and Lacis,
1996; Solmon et al., 2008; Pérez et al., 2006; Miller et al., 2014). Dust particles perturb the surface air
temperature through their radiative effect at TOA, can increase the atmospheric stability (e.g., Zhao et
al. 2011) and might affect precipitation at the global and regional scale (Solmon et al., 2008; Xian, 2008;
Vinoj et al., 2014; Miller et al., 2014 and references therein).
All models show that the effect of mineral dust on climate has a great sensitivity to the shortwave ab-
sorption properties of mineral dust (Lau et al., 2009; Loeb and Su, 2010; Ming et al., 2010; Perlwitz and
Miller, 2010). Absorption by mineral dust started receiving a great deal of interest in the last ten years
or so, when spaceborne and ground-based remote sensing studies (Dubovik et al., 2002; Colarco et al.,
2002; Sinyuk et al., 2003) suggested that mineral dust was less absorbing that it had been indicated by
in situ observations (e.g. Patterson et al., 1977; Haywood et al., 2001), particularly at wavelengths below
600 nm. Balkanski et al. (2007) showed that lowering the dust absorption properties to an extent that
reconcile them both with the remote-sensing observations and the state-of-knowledge of the mineralog-
ical composition, allowed calculating clear-sky dust shortwave radiative effect in agreement with satel-
lite-based observations. A significant body of observations have been performed in quantifying the
shortwave light-absorbing properties of mineral dust, by direct measurements (Alfaro et al., 2004; Linke
et al., 2006; Osborne et al., 2008; McConnell et al., 2008; Derimian et al., 2008; Yang et al., 2009;
Müller et al., 2009; Petzold et al., 2009; Formenti et al., 2011; Moosmüller et al., 2012; Wagner et al.,
2012; Ryder al., 2013a; Utry et al., 2015; Denjean et al., 2015c; 2016), and indirectly, by quantifying
the amount and the speciation of the light-absorbing compounds in mineral dust, principally iron oxides
(Lafon et al., 2004; 2006; Lazaro et al., 2008; Derimian et al., 2008; Zhang et al., 2008; Kandler et al.,
2007; 2009; 2011; Formenti et al., 2014a; 2014b).
However, existing data are often limited to a single wavelength, which moreover are not identical for all
experiments. Also, frequently they do not represent the possible regional variability of the dust absorp-
tion, either because they are obtained from field measurements integrating the contributions of different





source regions, or conversely, by laboratory investigation targeting samples from a limited number of
locations. This might lead to biases. Indeed, iron oxides in mineral dust, mostly in the form of hematite
($Fe_2O_3$) and goethite ($Fe(O)OH$), have specific absorption bands in the UV-VIS spectrum (Bédidi and
Cervelle, 1993), and have a variable content depending on the soil mineralogy of the source regions
(Journet et al., 2014).
Henceforth, in this study we present a new evaluation of the ultraviolet to near-infrared (375-850 nm)
light-absorbing properties of mineral dust by studying the size-segregated mass absorption efficiency
(MAE, units of $m^2\,g^{-1}$) and its spectral dependence, largely used in climate models to calculate the direct
radiative effect of aerosols. Experiments on twelve aerosol samples generated from natural parent top
soils from various source regions worldwide have been conducted with a large atmospheric simulation
chamber.
**2. Instruments and methods**
At a given wavelength λ, the mass absorption efficiency (MAE, units of $m^2\,g^{-1}$) is defined as the ratio of
the aerosol light-absorption coefficient $b_{abs}(\lambda)$ (units of $m^{-1}$) and its mass concentration (in $\mu g\,m^{-3}$)

$$MAE(\lambda) = \frac{b_{abs}(\lambda)}{Mass\ Conc} \qquad (1)$$


MAE values for mineral dust aerosol are expressed in.
The spectral dependence of the aerosol absorption coefficient $b_{abs}(\lambda)$ is described by the power-law
relationship

$$b_{abs}(\lambda) \sim \lambda^{-AAE} \qquad (2)$$


where the AAE is the Ångström Absorption Exponent, representing the negative slope of $b_{abs}(\lambda)$ in a
log-log plot (Moosmüller et al., 2009)





$$AAE = -\frac{d\ln(b_{abs}(\lambda))}{d\ln(\lambda)} \qquad (3)$$



### 2.1. The CESAM simulation chamber

Experiments in this work have been performed in the 4.2 m$^3$ stainless-steel CESAM (French acronym for Experimental Multiphasic Atmospheric Simulation Chamber) simulation chamber (Wang et al., 2011). The CESAM chamber has been extensively used in recent years to simulate, at sub and super-saturated conditions, the formation and properties of aerosols at concentration levels comparable to those encountered in the atmosphere (Denjean et al., 2015a; 2015b; Bregonzio-Rozier et al., 2015; 2016; Di Biagio et al., 2014; 2016).

CESAM is a multi-instrumented platform, equipped with twelve circular flanges to support its analytical environment. Basic instrumentation comprises sensors to measure the temperature, pressure and relative humidity within the chamber (two manometers MKS Baratrons (MKS, 622A and MKS, 626A) and a HMP234 Vaisala® humidity and temperature sensor). The particle size distribution is routinely meas-ured by a combination of (i) a scanning mobility particle sizer (SMPS, mobility diameter range 0.02–0.88 µm), composed of a Differential Mobility Analyzer (DMA, TSI Inc. Model 3080) and a Conden-sation Particle Counter (CPC, TSI Inc. Model 3772); (ii) a SkyGrimm optical particle counter (Grimm Inc., model 1.129, optical equivalent diameter range 0.25–32 µm); and (iii) a WELAS optical particle counter (PALAS, model 2000, optical equivalent diameter range 0.5–47 µm). Full details of operations and data treatment of the particle counters are provided in Di Biagio et al. (2016).

### 2.2. Filter sampling

Three filter samples per top soil sample were collected on different types of substrate based on the anal-ysis to perform. Sampling dedicated to the determination of the aerosol mass concentration by gravimet-ric analysis and the measurement of the absorption coefficients by optical analysis was performed on 47-mm quartz membranes (Pall Tissuquartz™, 2500 QAT-UP). Two samples were collected in parallel. The first quartz membrane sample ("total") was collected without a dedicated size cut-off using an in-house built stainless steel sampler operated at 5 L min$^{-1}$. However, as detailed in Di Biagio et al. (2016), the length of the sampling line from the intake point in the chamber to the filter entrance was 50 cm, yielding with a 50% cut-off of the transmission efficiency at 10.6 µm in particle aerodynamic diameter. This fraction is therefore indicated as PM$_{10.6}$ in the forthcoming discussion. The second quartz membrane





sample was collected using a 4-stage DEKATI impactor operated at the flow rate of 10 L min$^{-1}$ to select
the aerosol fraction of particles with aerodynamic diameter smaller than 2.5 µm, indicated as PM$_{2.5}$ here
forth. Sampling for the analysis of the iron oxide content was performed on polycarbonate filters (47-
mm Nuclepore, Whatman; pore size of 0.4 µm) using the same sample holder than used for the total
quartz filters, and therefore referring to the PM$_{10.6}$ mass fraction. Samples were collected at a flow rate
of 6 L min$^{-1}$. All flow rates were monitored by a thermal mass flow meter (TSI Inc., model 4140). These
samples were also used to determine the elemental composition (including Fe) and the fraction of iron
oxides in the total mass.
**2.3. The Multi-Wavelength Absorbance Analyzer (MWAA)**
The aerosol absorption coefficient b$_{abs}$(λ) at 5 wavelengths (λ = 375, 407, 532, 635 and 850 nm) was
measured by *in situ* analysis of the quartz filter samples using the Multi-Wavelength Absorbance Ana-
lyzer (MWAA), described in detail in Massabò et al. (2013; 2015).
The MWAA performs a non-destructive scan of the quartz filters on 64 different points, each ~ 1 mm$^2$
wide. It measures the light transmission through the filter as well as backscattering at two different angles
(125° and 165°). This is necessary to constrain the multiple scattering effects occurring within the par-
ticle-filter system. Measurements are used as input of a radiative transfer model (Hänel, 1987; 1994) as
implemented by Petzold and Schölinner (2004) for the Multi-Angle Absorption Photometry (MAAP)
measurements. In this model, a two stream approximation is applied (Coakley and Chylek, 1975), in
which the fractions of hemispherical backscattered radiation with respect to the total scattering for col-
limated and diffuse incident radiation are approximated on the basis of the Henyey-Greenstein scattering
phase function (Hänel, 1987). This approximation assumes a wavelength-independent asymmetry pa-
rameter (g) set to 0.75, appropriate for mineral dust (Formenti et al., 2011; Ryder et al., 2013b). The
total uncertainty, including the effects of photon counting and the deposit inhomogeneity, on the absorp-
tion coefficient measurement is estimated at 8%.
**2.4. Gravimetric analysis**
The aerosol mass deposited on filter (µg) was obtained by weighing the quartz filter before and after
sampling, after a period of 48 hours of conditioning in a room with controlled atmospheric conditions
(temperature, T ~ 20 ± 1 °C; relative humidity, RH ~ 50 ± 5%). Weighting is performed with an analyt-
ical balance (Sartorius model MC5, precision of 1 µg), and repeated three times to control the statistical
variability of the measurement. Electrostatic effects are removed by exposing the filters, prior weighing,



to a de-ionizer. The error on the measured mass is estimated at 10 μg, including the repetition variability.
The aerosol mass concentration (μg m$^{-3}$) is obtained by dividing the mass deposited on filter to the total
volume of sampled air (m$^3$) obtained from the mass flowmeter measurements.
**2.5. Dust composition measurements**
**2.5.1. Elemental composition**
Elemental concentrations for the major constituents of mineral dust (Na, Mg, Al, Si, P, S, Cl, K, Ca, Fe,
Ti, Mn) were obtained by a Wavelength Dispersive X-ray fluorescence (WD-XRF) of the Nuclepore
filters using a PW-2404 spectrometer by Panalytical. Excitation X-rays are produced by a Coolidge tube
($I_{max}$ = 125 mA, $V_{max}$ = 60 kV) with a Rh anode; primary X-ray spectrum can be controlled by inserting
filters (Al, at different thickness) between the anode and the sample. Each element was analyzed three
times, with specific conditions (voltage, tube filter, collimator, analyzing crystal and detector). Data
collection was controlled by the SuperQ software provided with the instrument. The elemental mass
thickness (μg cm$^{-2}$), that is, the analyzed elemental mass per unit surface, was obtained by comparing
the elemental yields with a sensitivity curve measured in the same geometry on a set of certified mono-
or bi-elemental thin layer standards by Micromatter Inc. The certified uncertainty on the standard deposit
(± 5%) determines the lower limit on the uncertainty on the measured elemental concentrations, which
range between 8% and 10% depending on the considered element. Thanks to the uniformity of the aer-
osol deposit on the filters, the atmospheric elemental concentrations (μg m$^{-3}$) were calculated by multi-
plying the analyzed elemental mass thickness by the ratio between the collection and analyzed surfaces
of each sample (41 and 22 mm, respectively), then divided by the total sampled volume (m$^3$). Finally,
concentrations of light-weight elements (atomic number Z < 19) were corrected for the underestimation
induced by the self-absorption of the emitted soft X-rays inside aerosol particles according to Formenti
et al. (2011).
Additional XRF analysis of the quartz filters has been performed both in the PM$_{10.6}$ and the PM$_{2.5}$ frac-
tions, so to verify the absence of biases between the experiments dedicated to the determination of par-
ticle composition to those where the optical properties where measured.
**2.6.2. Iron oxide content**
The content and the mineralogical speciation of iron oxides, also defined as free-iron, that is, the fraction
of iron which is not in the crystal lattice of silicates (Karickhoff and Bailey, 1973), was determined by





XANES (X-ray absorption near-edge structure) in the Fe K-range (K$_\alpha$, 7112 eV) at the SAMBA (Spec-
troscopies Applied to Materials based on Absorption) beamline at the SOLEIL synchrotron facility in
Saclay, France (Briois et al., 2011). The position and shape of the K pre-edge and edge peaks were
analyzed as they depend on the oxidation state of iron and the atomic positions of the neighboring ions,
mostly O$^+$ and OH$^-$.
As in Formenti et al. (2014b), samples were mounted in an external setup mode. A Si(220) double-
crystal monochromator was used to produce a monochromatic X-ray beam, which was 3000 x 250 μm$^2$
in size at focal point. The energy range was scanned from 6850 eV to 7800 eV at a step resolution
varying between 0.2 eV in proximity to the Fe-K absorption edge (at 7112 eV) to 2 eV in the extended
range. Samples were analyzed in fluorescence mode without prior preparation. One scan acquisition
lasted approximately 30 minutes, and was repeated three times to improve the signal-to-noise ratio.
The same analytical protocol was applied to five standards of Fe(III)-bearing minerals (**Table 1**), includ-
ing iron oxides (hematite, goethite) and silicates (illite, montmorillonite, nontronite). The standard spec-
tra were used to deconvolute the dust sample spectra to quantify the mineralogical status of iron. The
linear deconvolution, performed the Athena IFEFFIT freeware analysis program (Ravel and Newville,
2005), provided with the proportionality factors $\alpha_i$ representing the mass fraction of elemental iron to be
assigned to the $i$-th standard mineral. In particular, the values of $\alpha_{hem}$ and $\alpha_{goe}$ represent the mass frac-
tions of elemental iron that can be attributed to hematite and goethite, and $\alpha_{Fe\ ox}$ ($\alpha_{hem} + \alpha_{goe}$), the mass
fraction of elemental iron that can be attributed to iron oxides.
**2.6.3. Calculation of the iron oxide content**
The measured elemental concentrations obtained by X-ray fluorescence are expressed in the form of
elemental oxides and summed to estimate the total mineral dust mass concentration MC$_{dust}$ according to
the equation from Lide (1992)

$$[MC_{dust}]=1.12\times\left\{\begin{matrix}1.658[Mg]+1.889[Al]+2.139[Si]+1.399[Ca]+1.668[Ti]+1.582[Mn]\\ +(0.5\times1.286+0.5\times1.429+0.47\times1.204)[Fe]\end{matrix}\right\} \qquad (4)$$

The relative uncertainty on $MC_{dust}$, estimated from the analytical error on the measured concentrations,
does not exceed 6%.





The fractional mass ratio (in percent) of elemental iron ($MR_{Fe\%}$) with respect to the total dust mass con-
centration $MC_{dust}$ is then calculated as

$$MR_{Fe\,\%} = \frac{[Fe]}{[MC_{Dust}]} \times 100 \qquad (5)$$


The mass concentration of iron oxides or free-iron ($MC_{Fe\,ox}$), representing the fraction of elemental iron
in the form of hematite and goethite ($Fe_2O_3$ and $FeOOH$, respectively), is equal to

$$MC_{Fe\,ox} = MC_{hem} + MC_{goe} \qquad (6)$$


where $MC_{hem}$ and $MC_{goe}$ are the total masses of hematite and goethite. These can be calculated from the
values $\alpha_{hem}$ and $\alpha_{goe}$ from XANES analysis, which represent the mass fractions of elemental iron at-
tributed to hematite and goethite, as

$$MC_{hem} = \frac{\alpha_{hem} \, x \, [Fe]}{0.70} \qquad (7.a)$$

$$MC_{goe} = \frac{\alpha_{goe} \, x \, [Fe]}{0.63} \qquad (7.b)$$


where the values of 0.70 and 0.63 represent the mass molar fractions of Fe in hematite and goethite,
respectively. The relative errors on $MC_{hem}$ and $MC_{goe}$ are obtained from the uncertainties on values $\alpha_{hem}$
and $\alpha_{goe}$ from XANES analysis (less than 10%).
The mass ratio of iron oxides ($MR_{Fe\,ox\%}$) with respect to the total dust mass can then be calculated as

$$MR_{Fe\,ox\%} = MC_{Fe\,ox} \times MR_{Fe\,\%} \qquad (8)$$





### 3. Experimental protocol

At the beginning of each experiment, the chamber was evacuated by to $10^{-4}$-$10^{-5}$ hPa. Then, the reactor was filled with a mixture of 80% $N_2$ and 20% $O_2$ at a pressure slightly exceeding the current atmospheric pressure, in order to avoid contamination from ambient air. Experiments were conducted at ambient temperature and at a relative humidity <2%. As in Di Biagio et al. (2014; 2016), dust aerosols were generated by mechanical shaking of the soils, previously sieved to < 1000 μm and dried at 100 °C for about 1 h to remove any residual humidity. About 15 g of soil was placed in a Buchner flask and shaken for about 30 min at 100 Hz by means of a sieve shaker (Retsch AS200). The dust particles produced was then injected in the chamber by flushing the flask with $N_2$ at 10 L min$^{-1}$ for about 10-15 min, whilst continuing shaking the soil.

Dust was injected for about 10-15 minutes, and left suspended in the chamber for approximately 120 min thanks to the 4-wheel fan located in the bottom of the chamber body. Previous measurements at the top and bottom of the chamber showed that the fan ensures a homogeneous distribution of the dust starting approximately 10 minutes after the end of the injection (Di Biagio et al., 2014).

To compensate for the air extracted from the chamber by sampling, a particle-free flow of $N_2/O_2$, regulated in real time as a function of the total volume of sampled air, was re-injected in the chamber. To avoid excessive dilution the flow was limited to 20 L min$^{-1}$. Two experiments per soil type were conducted: a first experiment for sampling on the nuclepore polycarbonate filters (determination of the elemental composition and the iron oxide fraction) and *in situ* measurements of the infrared optical constants (Di Biagio et al., 2016), and a second experiment on total quartz filter and impactor for the study of dust MAE presented in this paper.

**Figure 1** illustrates as typical example the time series of the aerosol mass concentration during the two experiments conducted for the Libyan sample. The comparison demonstrates the repeatability of the dust concentrations, both in absolute values and in temporal dynamics. It also shows that the mass concentrations decreased very rapidly by gravitational settling within the first 30 minutes of the experiment (see also the discussion in Di Biagio et al. (2016)), after which concentrations only decrease by dilution. The filter sampling was started after this transient phase, and then continued through the end of the experiments, in order to collect enough dust on filter for the chemical analysis. Blank samples were collected before the start of the experiments by placing the loaded filter holders in line with the chamber and by flushing them for a few seconds with air coming from the chamber.





At the end of each experimental series with a given soil sample, the chamber was manually cleaned in
order to remove carry-over caused by resuspension of particles deposited to the walls. Background con-
centrations of aerosols in the chamber vary between 0.5 and 2.0 $\mu$g m$^{-3}$, i.e. a factor of 500 to 1000 below
the operating conditions.

**3. Results and discussion**

The geographical location of the soil collection sites is shown in **Figure 2**, whereas the coordinates are
summarized in **Table 2**. As discussed in Di Biagio et al. (2016), the selection of these soils and sediments
was governed by the need of representing the major arid and semi-arid regions worldwide, the need of
taking into account the mineralogical diversity of the soil composition at the global scale, and finally by
their availability in sufficient quantities for injection in the chamber. When doing so, we obtained a set
of twelve samples distributed worldwide but mostly in Northern and Western Africa (Libya, Algeria,
Mali, Bodélé) and the Middle East (Saudi Arabia and Kuwait). Individual samples from the Gobi desert
in Eastern Asia, the Namib Desert, the Strzelecki desert in Australia, the Patagonian deserts in South
America, and the Sonoran Desert in Arizona have also been investigated.

**3.1. Elemental composition and iron oxide content**

A total of 41 filters including 15 polycarbonate filters (12 samples and 3 blanks) and 25 quartz filters
(12 for the total fraction, 10 for the fine fraction and 3 blanks) were collected for analysis.
The dust mass concentration found by gravimetric analysis varied between 50 $\mu$g m$^{-3}$ and 5 mg m$^{-3}$, in
relatively good agreement with the dust mass concentration $MC_{dust}$ (Equation 4) based on X-ray fluores-
cence analysis: the slope of the linear regression between the calculated and the gravimetric values of
$MC_{dust}$ is 0.90 with R$^2$ = 0.86.
Di Biagio et al. (2016) show that clays are the most abundant mineral phases, together with quartz and
calcite, and that significant variability exists as function of the compositional heterogeneity of the parent
soils. Here we use the Fe/Ca and Si/Al elemental ratios obtained from X-ray Fluorescence analysis to
discriminate the origin of used dust samples. These ratios have been extensively used in the past to
discriminate the origin of African dust samples collected in the field (Chiapello et al., 1997; Formenti et
al., 2011; Formenti et al., 2014a). The values obtained during our experiments are reported in **Table 3.**
There is a very good correspondence between the values obtained for the Mali, Libya, Mauritania and
(to a lesser extent) Morocco experiments to values found in environmental aerosol samples. The only



major difference is observed for the Bodélé experiment, for which the Fe/Ca ratio is enriched by a factor
of 6 with respect to the field observations (Formenti et al., 2011; Formenti et al., 2014a). This could
reflect the fact that the Bodélé aerosol in the chamber is generated from a sediment sample and not from
a soil. As a matter of fact, the Bodélé sediment sample is constituted by a very fine powder which be-
comes very easily airborne. Henceforth, and contrary to the soil samples, this powder is likely to be
injected in the chamber with little or no size fractionation. As a consequence, the composition of the
aerosol collected in the chamber could reflect more that of the parent sedimentary soil than not the other
samples. On the other hand, Bristow et al. (2010) and Moskowitz et al. (2016) show that the iron content
and speciation of the Bodélé sediments is very heterogeneous at the source scale. For non-northern Af-
rican samples, the largest variability is observed for the Fe/Ca values, ranging from 0.1 to 8, whereas
the Si/Al ratio varied between 2.5 and 4.8. In this case, values are available in the literature for compar-
ison (e.g., Cornille et al., 1990; Reid et al., 1994; Eltayeb et al., 2001; Lafon et al., 2006; Shen et al.,
2007; Radhi et al., 2010; 2011; Formenti et al., 2011; 2014a; Scheuvens et al., 2013 and references
within). Values in the PM$_{2.5}$ fraction are very consistent with those obtained in the PM$_{10.6}$: their linear
correlation has a slope of 1.03 ($\pm$ 0.05) and a $R^2$ equal to 0.97, suggesting that the elemental composition
is relatively size-independent.
The mass fraction of total Fe ($MC_{Fe\%}$ from Equation 5), also reported in **Table 3**, ranged from 2.8 to
7.3%, values found for the Namibia and the Australia samples, respectively. This range is in good agree-
ment with values reported in the literature, taking into account that differences might be also due to the
method (direct measurement/calculation) and/or the size fraction over which the total dust mass concen-
tration is estimated (Chiapello et al., 1997; Reid et al., 1994; 2003; Derimian et al., 2008; Formenti et
al., 2001; 2011; 2014a; Scheuvens et al., 2013). The agreement of $MC_{Fe\%}$ values obtained by the XRF
analysis of polycarbonate filters (Equation 5) and those obtained from the XRF analysis of the quartz
filters, normalized to the measured gravimetric mass is well within 10% (that is, the percent error of
each estimate). An the exception are the samples of Bodélé and Algeria, for which the values obtained
from the analysis of the quartz filters are significantly lower than those obtained from the nuclepore
filters. We treat that as an additional source of error in the rest of the analysis, and add it to the total
uncertainty. In the PM$_{2.5}$, the content of iron is more variable, ranging from 4.4% (Morocco) to 33.6%
(Mali), showing a size dependence. A word of caution on this conclusion as the two estimates are not
necessarily consistent in the way that the total dust mass is estimated (from Equation 4 for the PM$_{10.6}$
fraction and by gravimetric weighing on the PM$_{2.5}$).



Finally, between 11 and 47% of iron in the samples can be attributed to iron oxides, in variable propor-
tions between hematite and goethite. The iron oxide fraction of total Fe in this study is on the lower end
of the range (36-72%) estimated for field dust samples of Saharan/Sahelian origin (Formenti et al.
2014b). The highest value of Formenti et al. (2014b), obtained for a sample of locally-emitted dust col-
lected at the Banizoumbou station in the African Sahel, is anyhow in excellent agreement with the value
of 62% obtained for an experiment (not shown here) using a soil collected in the same area. Likewise,
the proportions between hematite and goethite (not shown) are reproduced, showing that goethite is more
abundant than hematite. The mass fraction of iron oxides ($MR_{Fe\ ox\%}$), estimated from Equation 8 and
shown in Table 3, ranges between 0.7% (Kuwait) to 3.6% (Australia), which is in the range of available
field estimates (Formenti et al., 2014a; Moskowitz et al., 2016). For China, our value of $MR_{Fe\ ox\%}$ is
lower by almost a factor of 3 in comparison with that obtained on the same dust sample by Alfaro et al.
(2004) (0.9% against 2.8%), whereas on a sample from Niger (however not considered in this study) our
estimates and that by Alfaro et al. (2004) perfectly agree (5.8%). A possible underestimate of the iron
oxide fraction for samples other than those from the Sahara-Sahel area could be due to the fact that -
opposite to the experience of Formenti et al. (2014b) - the linear deconvolutions of the XANES spectra
were not always satisfactory (see Figure S1 in the supplementary). This resulted in a significant residual
between the observed and fitted XANES spectra. Indeed, the mineralogical reference for hematite is
obtained from a soil from Niger (Table 1) and might not be fully suitable for representing aerosols of
different origins. Additional differences could arise from differences in the size distributions of the gen-
erated aerosol. As a matter of fact, the number fraction of particles in the size classes above 0.5 µm in
diameter are different in the dust aerosol generated in the Alfaro et al. (2004) study with respect to ours.
In the study by Alfaro et al. (2004), the number fraction is lowest in the 0.5-0.7 size class and highest
between 1 and 5 µm. On the contrary, in our study the number fraction is lowest in the 1-2 µm size range
and highest between 0.5 and 0.7 µm. These differences could yield either to difference in the chemical
composition and/or to a difference in the total mass in the denominator of Equation 8.
**3.2. Spectral and size-variability of the mass absorption efficiency**
The spectral mass absorption efficiency (MAE) at 375, 407, 532, 635 and 850 nm for the $PM_{10.6}$ and the
$PM_{2.5}$ dust fractions are summarized in **Table 4** and displayed in **Figure 3**. Regardless of particle size,
the MAE values decrease with increasing wavelength (almost one order of magnitude between 375 and
850 nm), and display a larger variability at shorter wavelengths. The MAE values for the $PM_{10.6}$ range
from 37 ($\pm$ 3) $10^{-3}$ m$^2$ g$^{-1}$ to 135 ($\pm$ 11) $10^{-3}$ m$^2$ g$^{-1}$ at 375 nm, and from 1.3 ($\pm$ 0.1) $10^{-3}$ m$^2$ g$^{-1}$ to 15 ($\pm$ 1)



$10^{-3}$ $m^2$ $g^{-1}$ at 850 nm. Maxima are found for the Australia and Algeria samples, whereas the minima are
for Bodelé and Namibia, respectively at 375 and 850 nm. In the PM$_{2.5}$ fraction, the MAE values range
from 95 ($\pm$ 8) $10^{-3}$ $m^2$ $g^{-1}$ to 711 ($\pm$ 70) $10^{-3}$ $m^2$ $g^{-1}$ at 375 nm, and from 3.2 ($\pm$ 0.3) $10^{-3}$ $m^2$ $g^{-1}$ to 36 ($\pm$ 3)
$10^{-3}$ $m^2$ $g^{-1}$ at 850 nm. Maxima at both 375 and 850 nm are found for the Morocco sample, whereas the
minima are for Algeria and Namibia, respectively. The MAE values for mineral dust resulting from this
work are in relative good agreement with the estimates available in literature (Alfaro et al., 2004; Linke
et al., 2006; Yang et al., 2009; Denjean et al., 2016), reported in **Table 5**. For the China Ulah Buhn
sample, Alfaro et al. (2004) reported 69.1 $10^{-3}$ and 9.8 $10^{-3}$ $m^2$ $g^{-1}$ at 325 and 660 nm, respectively. The
former is lower than the value of 99 $10^{-3}$ $m^2$ $g^{-1}$ that we obtain by extrapolating our measurement at 375
nm. Likewise, our values for the Morocco sample are higher than reported by Linke et al. (2006) at 266
and 660 nm. Conversely, the agreement with the estimates of Yang et al. (2009) for mineral dust locally
re-suspended in Xianghe, near Beijing (China) is very good at all wavelengths between 375 and 880 nm.
As expected, the MAE values for mineral dust resulting from this work are almost one order of magni-
tude smaller than for other absorbing aerosols. For black carbon, MAE values are in the range of 6.5–
7.5 $m^2$ $g^{-1}$ at 850 nm (Bond and Bergstrom, 2006; Massabò et al., 2016), and vary in a linear way in-
versely with wavelength. For brown carbon, the reported MAE range between 2.3–7.0 $m^2$ $g^{-1}$ at 350 nm
(Chen and Bond, 2010; Kirchstetter et al., 2004; Massabò et al., 2016), 0.05–1.2 $m^2$ $g^{-1}$ at 440 nm (Wang
et al., 2016) and 0.08–0.72 $m^2$ $g^{-1}$ at 550 nm (Chen and Bond, 2010).
The analysis of **Table 4** indicates that, at every wavelength, the MAE values in the PM$_{2.5}$ fraction are
equal or higher than those for PM$_{10.6}$. The PM$_{2.5}$/PM$_{10.6}$ MAE ratios reach values of 6 for the Mali sam-
ple, but are mostly in the range 1.5-3 for the remaining aerosols. Values decrease with wavelength up to
635 nm, whereas at 850 nm they have values comparable to those at 375 nm. The observed size-depend-
ence of the MAE values is consistent with the expected behavior of light absorption of particles in the
Mie and geometric optical regimes that concern the two size fractions. Light-absorption of particles of
size smaller or equivalent to wavelength is proportional to their bulk volume, whereas for larger particles
absorption occurs on their surface only (Bohren and Huffmann, 1983). On the other hand, the size-
resolved measurements of Lafon et al. (2006) show that the proportion (by volume) of iron oxides might
be higher in the coarse than in the fine fraction, which would counteract the size-dependence behavior
of MAE. To validate the observations, we calculated the spectrally-resolved MAE values in the two size
fractions using the Mie code for homogeneous spherical particles (Bohren and Huffmann, 1983) and the
number size distribution estimated by (Di Biagio et al., 2016) and averaged over the duration of filter





sampling. We estimated the dust complex refractive index as a volume-weighted average of a non-ab-
sorbing dust fraction having the refractive index of kaolinite, dominant mineral in our samples (see Di
Biagio et al., 2016), from Egan et Hilgeman (1979) and an absorbing fraction estimated from the mass
fraction of iron oxides and having the refractive index of hematite (Bedidi and Cervelle, 1993). Results
of this calculation indicate that the observed size-behavior is well reproduced at all wavelengths, even
in the basic hypothesis that the mineralogical composition does not change with size. The only exception
is 850 nm, where at times, $PM_{2.5}/PM_{10.6}$ MAE ratio is much higher than expected theoretically. We
attribute that to the relatively high uncertainty affecting the absorbance measurements at this wavelength
where the signal-to-noise ratio is low. Indeed, the two sets of values (MAE in the $PM_{2.5}$ fraction and
MAE in the $PM_{10.6}$ fraction) are not statistically different according to a two-pair t-test (0.01 and 0.05
level of confidence), confirming that any attempt of differentiation the size-dependence at this wave-
length would require a stronger optical signal.
The analysis of the spectral dependence, using a power-law function fit as from Equation 2, provides
with the values of the Angstrom Absorption Exponent (AAE), also reported in **Table 4**. Contrary to the
MAE values, there is no statistically significant size-dependence of the AAE values, ranging from 2.5
(± 0.2) to 4.1 (± 0.3), with an average of 3.3 (± 0.7), for the $PM_{10.6}$ size fraction and between 2.6 (± 0.2)
and 5.1 (± 0.4), with an average of 3.5 (± 0.8), for the $PM_{2.5}$ fraction. Our values are in the range of those
published in the open literature (Fialho et al., 2005; Linke et al., 2006; Müller et al., 2009; Petzold et al.,
2009; Yang et al., 2009; Weinzierl et al., 2011; Moosmüller et al., 2012; Denjean et al., 2016), shown in
**Table 5**. AAE values close to 1.0 are found for urban aerosols where fossil fuels combustion is dominant,
while AAE values for brown carbon (BrC) from incomplete combustion are in the range 3.5-4.2 (Yang
et al., 2009; Chen et al., 2015; Massabò et al., 2016).
Finally, **Figure 4** shows correlations between MAE values in the $PM_{10.6}$ fraction (Figure 3.a) and in the
$PM_{2.5}$ fraction (Figure 3.b) and the estimated percent mass fraction of iron and iron oxides ($MC_{Fe\%}$ and
$MC_{Fe\ ox\%}$), respectively. Regardless of the size fraction, the correlation between the MAE values and the
percent mass of total elemental iron are satisfactory. Higher correlations are obtained at 375, 407 and
532 nm, and in the $PM_{2.5}$ fraction, where a linear correlation with $R^2$ up to 0.94 are obtained. Best cor-
relations are obtained when forcing the intercept to zero, indicating that elemental iron fully accounts
for the measured absorption. At these wavelengths, linear correlations with the mass fraction of iron
oxides are loose in the $PM_{10.6}$ mass fraction ($R^2$ up to 0.38-0.62), but again satisfactory in the $PM_{2.5}$
fraction ($R^2$ up to 0.83-0.99), where, whoever, one should keep in mind that they have been established





only indirectly by considering the ratio of iron oxides to elemental iron independent of size. At 660 and
850 nm, little or no robust correlation is obtained, often on very few data points and with very low MAE
values. It is noteworthy that, in both size fractions, the linear correlation yields a non-zero intercept is
obtained, indicating other minerals but iron oxides account for the measured absorption.
**4. Conclusive remarks**
In this paper, we reported new laboratory measurements of the shortwave mass absorption efficiency
(MAE) of mineral dust of different origin and as a function of size and wavelength in the 375-850 nm
range. Results have been obtained in the CESAM simulation chamber using generated mineral dust from
natural parent soils, and optical and gravimetric analysis on extracted samples.
Our results can be summarized as follows: at 375 nm, the MAE values are lower for the $PM_{10.6}$ mass
fraction (range 37-135 $10^{-3}$ $m^2$ $g^{-1}$) than for the $PM_{2.5}$ (range 95-711 $10^{-3}$ $m^2$ $g^{-1}$), and vary opposite to
wavelength as $\lambda^{-AAE}$, where AAE (Angstrom Absorption Exponent) averages between 3.3-3.5 regardless
of size fraction. These results deserve some conclusive comments:
•    The size-dependence, yielding significantly higher MAE values in the fine fraction ($PM_{2.5}$) than

for the $PM_{10.6}$ aerosol, indicates that light-absorption by mineral dust can be important even dur-

ing atmospheric transport over heavy polluted regions, when dust concentrations are significantly

lower than at emission. This can be shown by comparing the aerosol absorption optical depth

(AAOD) at 440 nm for China, a well-known mixing region of mineral dust and pollution (e.g.,

Yang et al., 2009; Laskin et al., 2014; Wang et al., 2013). Laskin et al. (2014) reports that the

average AAOD in the area is of the order of 0.1, for carbonaceous absorbing aerosols (sum of

black and brown carbon). This is lower or comparable to the AAOD of 0.17 and 0.11 at 407 nm

(fine and total mass fractions, respectively) that we obtain by a simple calculation (AAOD =

MAE x $MC_{dust}$ x H), where MAE are the values estimated in this study, MC the dust mass con-

centrations typically observed in the area during dust storms (Sun et al., 2001), and H a scale

height factor of 1 km.

•    The spectral variability of the dust MAE values, represented by the AAE parameter, is equal in

the $PM_{2.5}$ and $PM_{10.6}$ mass fractions. This suggests that, for a given size distribution, the possible

variation of dust composition with size do not affect in a significant way the spectral behavior of

the absorption properties. Our average value for AAE is 3.3 ± 0.7, higher than for black carbon,

but in the same range than light-absorbing organic (brown) carbon. As a result, depending on the





environment, there can be some ambiguity in apportioning the AAOD based on spectral depend-
ence. Bahadur et al. (2012) and Chung et al. (2012) couple the AAE and the spectral dependence
of the total AOD and/or its scattering fraction only to overcome this problem. Still, Bahadur et
al. (2012) show that there is an overlap in the scatterplots of the spectral dependence of the scat-
tering and absorption fractions of the AOD based on analysis of ground-based remote sensing
data for mineral dust, urban and non-urban fossil fuel over California. A closer look to observa-
tions in mixing areas where biomass burning have different chemical composition and/or mineral
dust has heavy loadings should be given in order to generalize the clear separation observed in
the spectral dependences of mineral dust and biomass burning (Bahadur et al., 2012). This aspect
is relevant to the development of remote sensing of light-absorption aerosols from space, and
their assimilation in climate models (Torres et al., 2007; Buchard et al., 2015; Hammer et al.,
2016).

• There is an important sample-to-sample variability in our dataset of MAE values for mineral dust
aerosols. At 532 nm, our estimated MAE average at $34 \pm 14$ m$^2$ g$^{-1}$ and $78 \pm 70$ m$^2$ g$^{-1}$ in the
PM$_{10.6}$ and PM$_{2.5}$ mass fractions, respectively. Figure 3, showing the correlation with the esti-
mated mass fraction of elemental iron and iron oxides, suggests that this variability could be
related to the regional differences of the mineralogical composition of the parent soils. These
observations lead to different considerations. To start with, our study reinforces the need for
regionally-resolved representation of the light-absorption properties of mineral dust in order to
improve the representation of its effect on climate. As a matter of fact, the natural variability of
the absorption properties that we obtain from our study is in the range 50-100%, even when we
limit ourselves to smaller spatial scales, for example those of north Africa (samples from Libya,
Algeria, Mali and Bodélé). This is far above the $\pm$ 5% sensitivity factor used by Solmon et al.
(2008) to vary the single scattering albedo (as a proxy of absorption) of mineral dust over western
Africa, and to show how this could drastically change the climate response in the region.
The question is then "how to represent this regional variability?" As Moosmüller et al. (2012),
we found that elemental iron is a very good proxy for the MAE, especially in the PM$_{2.5}$ fraction,
where iron-bearing absorbing minerals (hematite, goethite, illite, smectite clays) would be more
concentrated. In the coarse fraction, Ca-rich minerals, quartz and feldspars could also play a role,
and that could result in the observed lowered correlation (although adding a term proportional to
elemental Ca does not ameliorate the result in the present study). The correlation of the spectral





MAE values with the iron oxide fraction is satisfactory but rather noisy, also owing to some
uncertainty in the quantification of iron oxides from X-Ray Absorption measurements. In this
case, the intercept is significantly different from zero, again indicating that a small but clear
fraction of absorption is due to minerals other than iron oxides. There are contrasting results on
this topic: Alfaro et al. (2004) found an excellent correlation between MAE and the iron oxide
content, whereas Klaver et al. (2011) found that the single scattering albedo (representing the
capacity of an aerosol population to absorb light with respect to extinction) was almost inde-
pendent on the mass fraction of iron oxides. Moosmüller et al. (2012) disagreed, pointing out to
the uncertainty in the correction procedure of the measurement of absorption by Klaver et al.
(2011). As a matter of fact, Klaver et al. (2011) and Alfaro et al. (2004) used the same correction
procedure. It is more likely that the lack of correlation found in Klaver et al. (2011) is due to the
fact that other minerals than iron oxides contribute to absorption, in particular at their working
wavelength (567 nm), where the absorption efficiency of iron oxides starts to weaken. Clearly,
the linear correlation between elemental iron in mineral dust and its light-absorption properties
could ease the application and validation of climate models that now starting including the rep-
resentation of the mineralogy (Perlwitz et al., 2015a; 2015b; Scanza et al., 2015). Also, they
would facilitate detecting source regions based on remote sensing of dust absorption in the UV-
VIS spectral region (e.g., Hsu et al., 2004). However, such a quantitative relationship cannot
uniquely determined from these studies, including the present one, which use different ways of
estimating elemental iron, iron oxides and the total dust mass. A more robust estimate should be
obtained by estimating the imaginary parts of the complex refractive indices associated to these
measurements of absorption, and their dependence on the mineralogical composition.
**Author contributions**
L. Caponi, P. Formenti, D. Massabò, P. Prati, C. Di Biagio, and J. F. Doussin designed the chamber
experiments and discussed the results. L. Caponi and C. Di Biagio realized the experiments with contri-
butions by M. Cazaunau, E. Pangui, P. Formenti, and J.F. Doussin. L. Caponi, D. Massabò and P. For-
menti performed the full data analysis with contributions by C. Di Biagio, P. Prati and J.F. Doussin. L.
Caponi, P. Formenti and S. Chevaillier performed the XRF measurements. P. Formenti and G. Landrot
performed the XAS measurements. D. Massabò performed the MWAA and the gravimetric measure-
ments. M. O. Andreae, K. Kandler, T. Saeed, S. Piketh, D. Seibert, and E. Williams collected the soil



samples used for experiments. L. Caponi, P. Formenti, D. Massabò and P. Prati wrote the manuscript
with comments from all co-authors.
**Acknowledgements**
The RED-DUST project was supported by the French national programme LEFE/INSU, by the EC
within the I3 project "Integration of European Simulation Chambers for Investigating Atmospheric Pro-
cesses" (EUROCHAMP-2020, grant agreement n. 730997), by the Institut Pierre Simon Laplace (IPSL),
and by OSU-EFLUVE (Observatoire des Sciences de l'Univers-Enveloppes Fluides de la Ville à l'Ex-
obiologie) through dedicated research funding. C. Di Biagio was supported by the CNRS via the Labex
L-IPSL. M. O. Andreae was supported by the Max Society and by King Saud University. The mobility
of researchers between Italy and France was supported by the PICS programme MedMEx of the CNRS-
INSU. The authors acknowledge the CNRS-INSU for supporting CESAM as national facility. K. Kan-
dler acknowledges support from the Deutsche Forschungsgemeinschaft (DFG grant KA 2280/2-1).

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





**Table captions**
**Table 1.** Characteristics of the standards used for the quantification of the iron oxides in the XAS anal-
ysis.
**Table 2.** Summary of information on the soil samples used in this work.
**Table 3.** Chemical characterisation of the dust aerosols in $PM_{10.6}$ and $PM_{2.5}$ (in parentheses) size frac-
tions. Columns 3 and 4 report the Si/Al and Fe/Ca elemental ratios obtained from X-Ray Fluorescence
analysis. The uncertainty on each individual value is estimated to be 10%. Column 5 reports $MR_{Fe\%}$, the
fractional mass of elemental iron with respect to the total dust mass concentration (uncertainty 10%).
Column 5 reports $MR_{Fe\%}$, the mass fraction of iron oxides with respect to the total dust mass concentra-
tion (uncertainty 15%). For $PM_{2.5}$ the determination of the Si/Al ratio is impossible due to the composi-
tion of the filter medium.
**Table 4.** Mass absorption efficiency (MAE, $10^{-3}$ $m^2$ $g^{-1}$) and Ångström Absorption Exponent (AAE) in
the $PM_{10.6}$ and $PM_{2.5}$ size fractions. Absolute errors are in brackets.
**Table 5.** Mass absorption efficiency (MAE, $10^{-3}$ $m^2$ $g^{-1}$) and Ångström Absorption Exponent (AAE) of
literature data discussed in the paper

**Figure captions**
**Figure 1.** Time series of aerosol mass concentration in the chamber for the two companion experiments
(Libya sample). Experiment 1 (top panel) was dedicated to the determination of the chemical composi-
tion (including iron oxides) by sampling on polycarbonate filters. Experiment 2 (bottom panel) was
dedicated to the determination of the absorption optical properties by sampling on quartz filters.
**Figure 2.** Location (red star) of the soil and sediment samples used to generate dust aerosols.
**Figure 3.** Spectral dependence of the MAE values for the samples investigated in this study in the $PM_{10.6}$
(left) and in the $PM_{2.5}$ (right) mass fractions.
**Figure 4.** Illustration of the links between the MAE values and the dust chemical composition found in
this study. Left column, from top to bottom: linear regression between MAE values between 375 and
850 nm and the fraction of elemental iron with respect to the total dust mass ($MR_{Fe\%}$) in the $PM_{10.6}$
fraction; Middle column: same as left column but respect to the mass fraction of iron oxides to the total

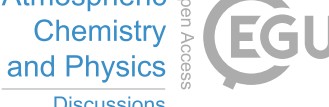

dust mass ($MR_{Fe\ ox\%}$) in the $PM_{10.6}$ size fraction; Right column: same as left column but in the $PM_{2.5}$ size
fraction.





**Table 1.** Characteristics of the standards used for the quantification of the iron oxides in the XAS anal-
ysis.

| Standard | Stoechiometric Formula | Origin |
|---|---|---|
| Illite of Puy | $(Si_{3.55}Al_{0.45})(Al_{1.27}Fe_{0.36}Mg_{0.44})O_{10}(OH)_2(Ca_{0.01}Na_{0.01}K_{0.53}X(I)_{0.12})$ | Puy, France |
| Goethite | $FeO\ OH$ | Minnesota |
| Hematite | $Fe_2O_3$ | Niger |
| Montmorillonite | $(Na,Ca)_{0,3}(Al,Mg)_2Si_4O10(OH)_2 \cdot n(H_2O)$ | Wyoming |
| Nontronite | $Na_{0.3}Fe_2(Si,Al)_4O10(OH)_2 \cdot nH2O$ | Pennsylvania |






**Table 2.** Geographical information on the soil samples used in this work.

| Geographical area | Sample | Desert area | Geographical coordinates |
|---|---|---|---|
| Sahara | Morocco | East of Ksar Sahli | 31.97°N, 3.28°W |
| | Libya | Sebha | 27.01°N, 14.50°E |
| | Algeria | Ti-n-Tekraouit | 23.95°N, 5.47°E |
| Sahel | Mali | Dar el Beida | 17.62°N, 4.29°W |
| | Bodélé | Bodélé depression | 17.23°N, 19.03°E |
| Middle East | Saudi Arabia | Nefud | 27.49°N, 41.98°E |
| | Kuwait | Kuwaiti | 29.42°N, 47.69°E |
| Southern Africa | Namibia | Namib | 21.24°S, 14.99°E |
| Eastern Asia | China | Gobi | 39.43°N, 105.67°E |
| North America | Arizona | Sonoran | 33.15 °N, 112.08°W |
| South America | Patagonia | Patagonia | 50.26°S, 71.50°W |
| Australia | Australia | Strzelecki | 31.33°S, 140.33°E |



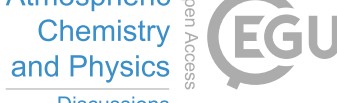

**Table 3.** Chemical characterisation of the dust aerosols in $PM_{10.6}$ and $PM_{2.5}$ (in parentheses) size frac-
tions. Columns 3 and 4 report the Si/Al and Fe/Ca elemental ratios obtained from X-Ray Fluorescence
analysis. The uncertainty on each individual value is estimated to be 10%. Column 5 reports $MR_{Fe\%}$, the
fractional mass of elemental iron with respect to the total dust mass concentration (uncertainty 10%).
Column 5 reports $MR_{Fe\ ox\%}$, the mass fraction of iron oxides with respect to the total dust mass concen-
tration (uncertainty 15%). For $PM_{2.5}$ the determination of the Si/Al ratio is impossible due to the com-
position of the filter medium.

| Geographical area | Sample | Si/Al | Fe/Ca | $MC_{Fe\%}$ | $MC_{Fe-ox\%}$ |
|---|---|---|---|---|---|
| Sahara | Morocco | 3.12 (---) | 0.24 (0.28) | 3.6 (4.4) | 1.4 (1.8) |
| | Libya | 2.11 (---) | 1.19 (1.12) | 5.2 (5.6) | 3.1 (3.4) |
| | Algeria | 2.51 (---) | 3.14 (4.19) | 6.6 (5.4) | 2.7 (2.2) |
| Sahel | Mali | 3.03 (---) | 2.99 (3.67) | 6.6 (33.6) | 3.7 (18.7) |
| | Bodélé | 5.65 (---) | 12.35 (----) | 4.1 (----) | 0.7 (----) |
| Middle East | Saudi Arabia | 2.95 (---) | 0.29 (0.27) | 3.8 (5.1) | 2.6 (3.5) |
| | Kuwait | 3.15 (---) | 0.89 (1.0) | 5.0 (13.6) | 1.5 (4.2) |
| Southern Africa | Namibia | 3.41 (---) | 0.11 (0.10) | 2.4 (6.9) | 1.1 (3.1) |
| Eastern Asia | China | 2.68 (---) | 0.77 (0.71) | 5.8 (13.6) | 0.9 (2.5) |
| North America | Arizona | 3.30 (---) | 0.95 (----) | 5.3 (----) | 1.5 (----) |
| South America | Patagonia | 4.80 (---) | 4.68 (4.64) | 5.1 (----) | 1.5 (---) |
| Australia | Australia | 2.65 (---) | 5.46 (4.86) | 7.2 (11.8) | 3.6 (5.9) |







**Table 4.** Mass absorption efficiency (MAE, $10^{-3}$ $m^2$ $g^{-1}$) and Ångström Absorption Exponent (AAE) in
the $PM_{10.6}$ and $PM_{2.5}$ size fractions. Absolute errors are in brackets.

| | | $PM_{10.6}$ | | | | | |
|---|---|---|---|---|---|---|---|
| **Geographical area** | **Sample** | **375 nm** | **407 nm** | **532 nm** | **635 nm** | **850 nm** | **AAE** |
| Sahara | Morocco | --- (---) | --- (---) | --- (---) | --- (---) | --- (---) | --- (---) |
| | Libya | 89 (11) | 75 (9) | 30 (5) | --- (---) | --- (---) | 3.2 (0.3) |
| | Algeria | 99 (10) | 80 (10) | 46 (7) | 16 (3) | 15 (3) | 2.5 (0.3) |
| Sahel | Mali | --- (---) | 103 (18) | 46 (12) | --- (---) | --- (---) | --- (---) |
| | Bodélé | 37 (4) | 25 (3) | 13 (2) | 6 (1) | 3 (1) | 3.3 (0.3) |
| Middle East | Saudi Arabia | 90 (9) | 79 (8) | 28 (3) | 6 (1) | 4 (1) | 4.1 (0.4) |
| | Kuwait | --- (---) | --- (---) | --- (---) | --- (---) | --- (---) | 2.8 (0.3) |
| Southern Africa | Namibia | 52 (7) | 49 (7) | 13 (3) | 5 (2) | 1 (2) | 4.7 (0.5) |
| Eastern Asia | China | 65 (8) | 58 (7) | 32 (4) | 8 (2) | 7 (2) | 3 (0.3) |
| North America | Arizona | 130 (15) | 99 (12) | 47 (7) | 21 (4) | 13 (4) | 3.1 (0.3) |
| South America | Patagonia | 102 (11) | 80 (9) | 29 (4) | 17 (2) | 10 (2) | 2.9 (0.3) |
| Australia | Australia | 135 (15) | 121 (13) | 55 (7) | 26 (4) | 14 (3) | 2.9 (0.3) |




| Geographical area | Sample | PM$_{2.5}$ | | | | | |
| --- | --- | --- | --- | --- | --- | --- | --- |
| | | 375 nm | 407 nm | 532 nm | 635 nm | 850 nm | AAE |
| Sahara | Morocco | 107 (13) | 88 (11) | 34 (6) | 14 (3) | 15 (4) | 2.6 (0.3) |
| | Libya | 132(17) | 103 (14) | 33 (7) | --- (---) | --- (---) | 4.1 (0.4) |
| | Algeria | 95(8) | 71 (11) | 37 (7) | 12 (5) | 12 (5) | 2.8 (0.3) |
| Sahel | Mali | 711 (141) | 621 (124) | 227 (78) | --- (---) | --- (---) | 3.4 (0.3) |
| | Bodélé | --- (---) | --- (---) | --- (---) | --- (---) | --- (---) | --- (---) |
| Middle East | Saudi Arabia | 153 (18) | 127 (15) | 42 (7) | 8 (4) | 6 (4) | 4.5 (0.5) |
| | Kuwait | 270 (100) | 324 (96) | --- (---) | 54 (52) | --- (---) | 3.4 (0.3) |
| Southern Africa | Namibia | 147 (36) | 131 (32) | 31 (21) | 6 (16) | 3 (15) | 5.1 (0.5) |
| Eastern Asia | China | 201 (30) | 176 (26) | 89 (17) | 14 (10) | 23 (10) | 3.2 (0.3) |
| North America | Arizona | --- (---) | --- (---) | --- (---) | --- (---) | --- (---) | --- (---) |
| South America | Patagonia | --- (---) | --- (---) | --- (---) | --- (---) | --- (---) | 2.9 (0.3) |
| Australia | Australia | 335 (39) | 288 (33) | 130 (19) | 57 (11) | 36 (9) | 2.9 (0.3) |




**Table 5.** Mass absorption efficiency (MAE, $10^{-3}$ m$^2$ g$^{-1}$) and Ångström Absorption Exponent (AAE) of
literature data discussed in the paper

| Geographical area | Sample | 266 nm | 325 nm | 428 nm | 532 nm | 660 nm | 880 nm | 1064 nm | AAE |
|---|---|---|---|---|---|---|---|---|---|
| Sahara | Morocco[*] | | | | | | | | 2.25–5.13 |
| | Morocco, PM$_{2.5}$[£] | | | | | | | | 2.0–6.5 |
| | Morocco, submicron[#] | 1100 | | | 60 | | | 30 | 4.2 |
| | Egypt, submicron[#] | 810 | | | 20 | | | | 5.3 |
| | Tunisia[$] | | 83 | | | 11 | | | |
| | Saharan, transported[μ] | | | | | | | | 2.9 ± 0.2 |
| | Saharan, transported (PM$_{10}$)[%] | | | 37 | 27[%%%] | 15[%%%%] | | | 2.9 |
| | Saharan, transported (PM$_1$)[%] | | | 60 | 40[%%%] | 30[%%%%] | | | 2.0 |
| Sahel | Niger[$] | | 124 | | | 19 | | | |
| Eastern Asia | China[$] | | 69 | | | 10 | | | |
| | China[&] | | 87[&&] | 50[&&&] | 27[&&&&] | 13 | 1 | | 3.8 |
| Arabian Peninsula, N/NE Africa, Central Asia | Various locations[@] | | | | | | | | 2.5-3.9 |

[*] Müller et al. (2008)
[£] Petzold et al. (2008)
[#] Linke et al. (2006)
[$] Alfaro et al. (2004)
[μ] Fialho et al. (2005)
[%] Denjean et al. (2016); [%%%] at 528 nm, [%%%%] at 652 nm
[&] Yang et al. (2009); [&&] at 375 nm, [&&&] at 470 nm, [&&&&] at 590 nm
[@] Mossmüller et al. (2012)





**Figure 1.** Time series of aerosol mass concentration in the chamber for the two companion experiments
(Libya sample). Experiment 1 (top panel) was dedicated to the determination of the chemical composi-
tion (including iron oxides) by sampling on polycarbonate filters. Experiment 2 (bottom panel) was
dedicated to the determination of the absorption optical properties by sampling on quartz filters.

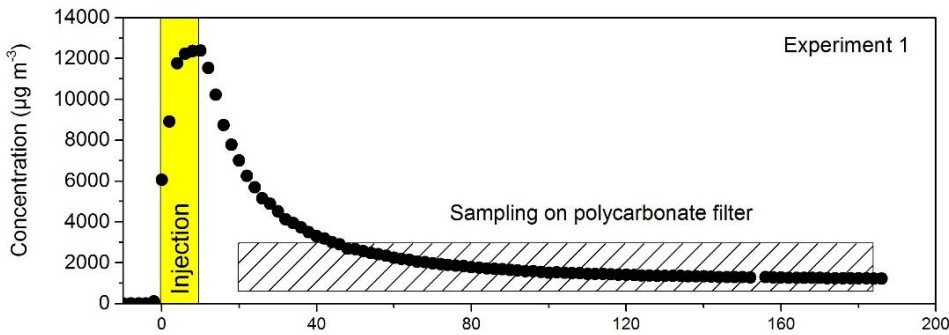

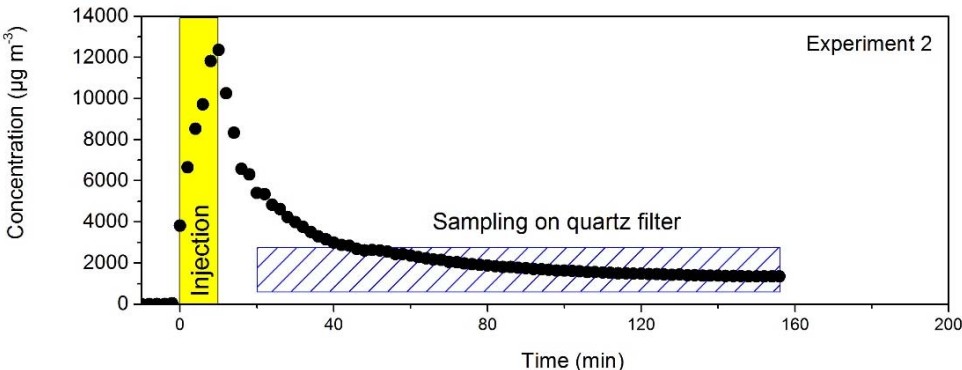


**Figure 2.** Location (red star) of the soil and sediment samples used to generate dust aerosols.





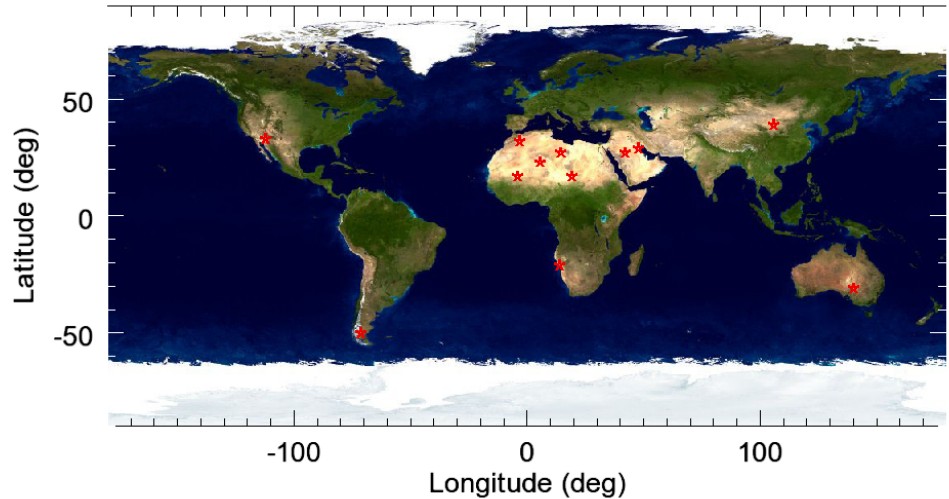






**Figure 3.** Spectral dependence of the MAE values for the samples investigated in this study in the PM$_{10.6}$
(left) and in the PM$_{2.5}$ (right) mass fractions.

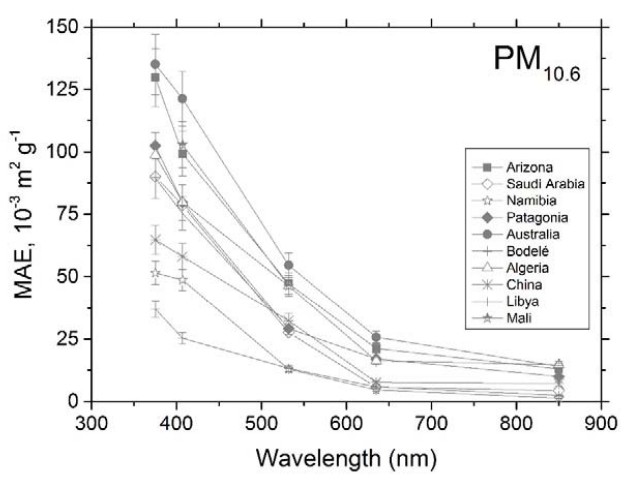


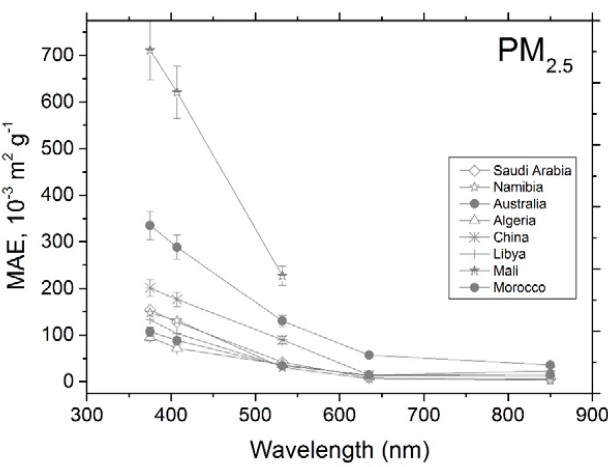




**Figure 4.** Illustration of the links between the MAE values and the dust chemical composition found in this study. Left column, from top to bottom: MAE values between 375 and 850 nm versus the fraction of elemental iron with respect to the total dust mass (MR$_{Fe\%}$) in the PM$_{10.6}$ fraction; Middle column: same as left column but versus the mass fraction of iron oxides to the total dust mass (MR$_{Fe\ ox\%}$) in the PM$_{10.6}$ size fraction; Right column: same as left column but in the PM$_{2.5}$ size fraction. The linear regression lines between MAE and MR$_{Fe\%}$ and MAE and MR$_{Fe\ Ox\%}$ are reported in each plot.

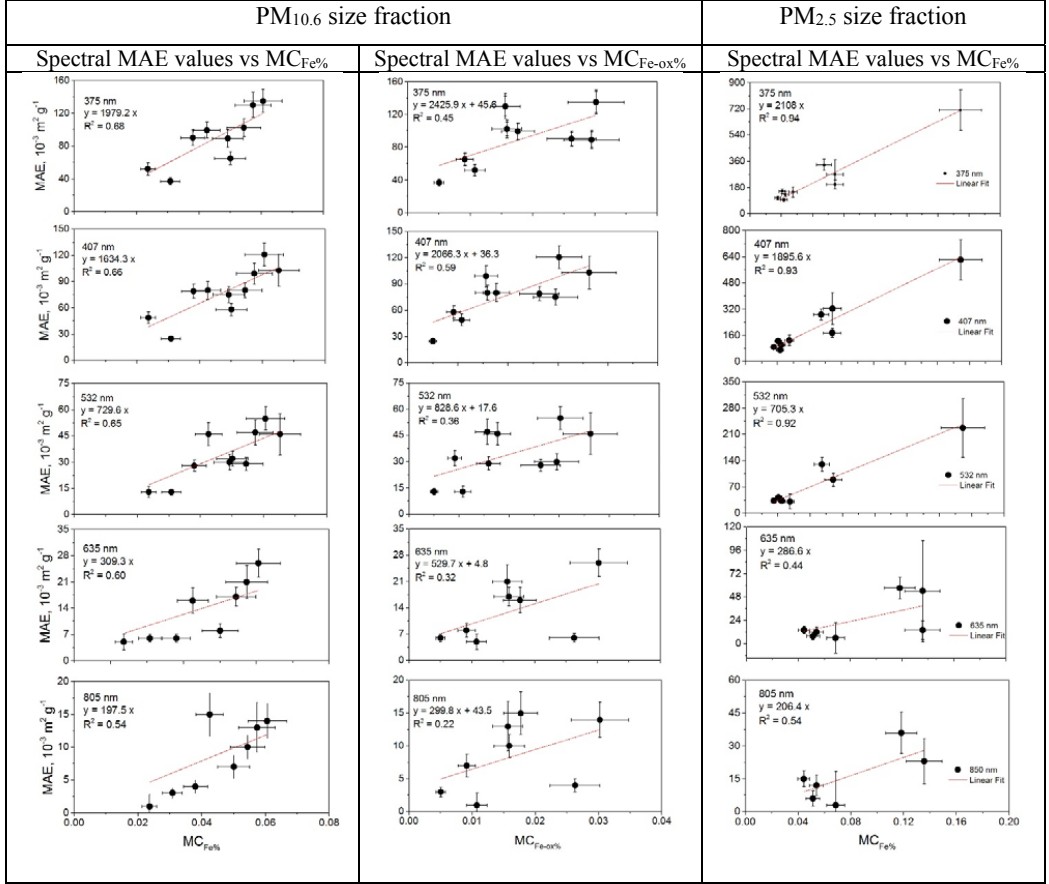