# Peer review of "Spectral- and size-resolved mass absorption efficiency of mineral dust aerosols in"

_Atmospheric Chemistry and Physics, 2017_

## Referee Comment (RC1) · Anonymous Referee #1 · 1 Feb 2017

The paper presents needed results on dust optical properties (mass absorption efficiency and absorption Angstrom exponent) for D<2.5um and D<10um particles, relating them to chemical composition. The samples analyzed are from 12 locations in northern Africa (5 samples), Namibia (1 sample), northern China (1 sample), the Middle East (2 samples), North America (1 sample), South America (1 sample) and Australia (1 sample). I have no substantial issues with the analysis or the paper. I recommend publication after addressing the minor points below:

1) In the Abstract and somewhere earlier in the paper (before the Results section) the

[Figure]

location where the samples were collected from should be presented. Some information about sample collection is also needed – i.e.: a) Was only one sample collected at each location, or were multiple samples collected then combined? b) Were the samples collected from locations know to be preferential sources for atmospheric dust or was the collection location just random? The latter point is important, since it's know that atmospheric dust comes from preferential locations.

2) Abstract, pg. 2, lines 33-34: "The size-independence of AAE suggests that, for a given size distribution, the possible variation of dust composition with size would not affect significantly the spectral behavior of shortwave absorption." Either that or the composition simply DIDN'T vary with size for this set of samples – so I'd reword this a bit.

3) pg. 2, line 39: Need to spell out AAOD this first time that you use it.

4) pg. 2, lines 40-41: "...which is relevant to the development of remote sensing of light-absorption aerosols from space ..." Not only from space! This approach has also been used extensively for AERONET (surface-based remote sensing) AAOD attribution.

5) pg. 3 line 56 contains a partial sentence ("In the solar spectrum (Boucher et al., 2013)").

6) pg. 6, lines 162-163: "The total uncertainty, including the effects of photon counting and the deposit inhomogeneity, on the absorption coefficient measurement is estimated at 8%." Is there a basis/reference for this?

7) pg. 7, line 170: It would be useful to give typical total masses and/or the fractional error/uncertainty in total aerosol gravimetric mass based on this error in filter mass.

8) pg. 8, line 223 equation: When I read this my immediate question was: "How does this compare to the measure of total gravimetric mass?". You answer this question (appropriately) in the results section but I still think it would be useful to add a note

here pointing to the fact the in your results discussion you found this agreed well with the total gravimetric mass.

9) pg. 10: Do you have any estimate / sense of how well the dust suspended by this "shaking" compares to the dust lofted by winds?

10) pg. 11, line 209: Mauritania is not listed as one of the sample site locations in Table 3. ?

11) pg. 11, lines 309-310 (an onward): The results here are said to agree well with that found for atmospheric aerosols in other studies, but the values in these studies is not given so this feels very hand-waving and unconvincing. Are you referring to the values given in Table 5? If so, please refer directly to them. If not, the comparison here needs to be more quantitative (discuss numbers from the literature vs. what is found here).

12) pg. 13, lines 353-354: If you have results for Niger why not show them?

13) pg. 14, lines 387-388: MAE doesn't vary linearly inversely with wavelength, it varies linearly inversely with the log of the wavelength (hence our ability to use the AAE relationship).

14) pg. 15, lines 429-434: A few things: a) "satisfactory" and "loose" are not quantitative terms, nor are they really appropriate for a scientific paper. What constitutes "satisfactory"? Best is to just give the correlations. b) The high correlation coefficients for PM2.5 are really driven by one high data point and so are probably not very robust.

15) pg. 16, lines 455-456: How can fine-mode-only AAOD be GREATER than total aerosol AAOD?

16) pg. 17, lines 487-488: As written this implies Solomon et al. varied SSA by 5%. For, say, SSA of 0.9, that they varied SSA by 0.045. That is, the co-albedo (or absorption) was varied by 45% (0.1+/-0.045). This is the proper comparison to make to the variation in MAE that you calculate.

17) Overall: Some editing is needed for language throughout. Here I list some that stood out to me – all small stuff but editing would help readability:

pg. 3, line 67-68: "in the last ten years or so" (too casual for scientific writing)

pg. 3, line 74: "A significant body of observations have been performed. . .."

pg. 8, lines 214-215: "The linear deconvolution, performed the Athena IFEFFIT free-ware analysis program (Ravel and Newville, 2005), provided with the proportionality factors alpha_i representing the mass fraction of elemental iron to be assigned to the i-th standard mineral." (I found this sentence nearly impossible to follow. . .)

pg. 10, line 253: "the chamber was evacuated by to" (delete "by")

pg. 10, line 259: "dust particles produced was" –> "dust particles produced were"

pg. 10, lines 279 "dust on filter for" –> "dust on the filter for"

pg. 10, line 280: "by placing the loaded filter holders": This reads as if you are placing LOADED FILTERS (vs holders with blank filters in them, which is what I assume you mean). Reword.

pg. 11, line 206: "the origin of used dust samples". I think this should be "the origin of our dust samples", yes?

pg. 12, line 315: "Henceforth, and contrary to the soil samples. . ."

pg. 12, lines 317-318: ". . .could reflect more that of the parent sedimentary soil than not the other samples."

pg. 12, line 335: "An the exception"

pg. 13, lines 353-354: "As a matter of fact, the number fraction of particles in the size classes above 0.5 $\mu$m in diameter are different in the dust aerosol generated in the Alfaro et al. (2004) study with respect to ours."

pg. 13, line 364: "On the contrary" –> "In contrast"

pg. 13, line 365: "These differences could yield either to difference in the . . ."

pg. 15, lines 415-417: "using a power-law function fit as from Equation 2, provides with the values of. . ."

pg. 16, line 448: "The size-dependence, yielding significantly higher MAE values. . ."

pg. 17, line 470: "A closer look to observations" –> "A closer look at observations"

pg. 17, line 478: "our estimated MAE average at" –> "our average MAE values are"

pg. 17, line 490: "As Moosmuller et al." –> "As in Moosmuller et al."

pg. 18, line 503-504: "pointing out to the" –> "pointing out the"

---

## Referee Comment (RC2) · Anonymous Referee #2 · 1 Mar 2017

GENERAL COMMENT

The manuscript presents important results from a carefully designed and conducted study on the light-absorbing properties of mineral dust from various origins. Dust samples collected at the different source regions have been re-suspended in an aerosol chamber and characterized with respect to microphysical, optical and chemical properties by state of the art methods. The study provides urgently needed knowledge on the multi-spectral light absorbing properties of mineral dust and for sure deserves publication in ACP. The manuscript is well structured, the methods are described in necessary detail and the referenced literature reflects the current state of knowledge. I recommend publication after the following minor revisions have been considered.

SPECIFIC REMARKS

1. In the experimental protocol section, the potential impact of gravitational settling on the re-suspended fraction of the dust samples is mentioned. Given the instrumentation list, the size distributions of the airborne dust samples were monitored during the runs of the experiments. It appears obvious to control the change of the size distribution during the experiment time in the chamber. Since the mass concentration in the chamber decreased very rapidly after injection, whereas the chemical composition of the dust samples was determined from bulk samples, it would be important to know if the airborne fraction sampled for the determination of optical properties features the same chemical properties as the bulk samples. At least a discussion of this potential source of uncertainties should be presented, along with a plot showing the change of the size distributions during the experiment time. The current analysis starts from the assumption that the dust bulk properties represent also the properties of the sampled airborne fractions. However, is this really justified?

2. In section 3.2, the variability of dust optical properties with particle size is discussed. The authors found no statistically significant size-dependence of the absorption Ångström exponent (AAE), whereas the absolute values of the mass absorption efficiencies (MAE) show large differences between the $PM_{2.5}$ and $PM_{10}$ fractions with larger values for the fine mode fraction. These findings imply that the relative chemical composition with respect to light-absorbing compounds does not change between the size fractions (similar AAE values), whereas the differences between the MAC values indicate that coarse mode particles contain more non-absorbing matter than fine mode particles (higher MAE values for smaller particles). This however, this is in contrast to the assumption that the chemical composition is uniformly distributed across the particle size distribution. Here, a detailed discussion is requested.

3. In section 4, it is discussed that the potential impact of light absorption by mineral dust may play an important role even after long range transport. Cited studies all refer to observations in China. However, there is another detailed study on this effect available for the pollution plume of Dakar mixing with mineral dust which also includes the variation of the AAE during mixing (Petzold et al., 2011). The authors may consider including this study.

MINOR COMMENTS

1. The list of references contains various references which are not cited in the manuscript. This should be checked, I found the following but there may be more:

Anderson et al., 1998, Andrews et al., 2006, Arnott et al., 2005, Collaud Coen et al., 2010, Petzold et al., 2013.

2. Line 56: The sentence seems to be incomplete.

3. Line 97 – 98: The basic unit of mass concentrations is $g\,m^{-3}$. Using this unit, then the unit of the combined property MAE is $m^2\,g^{-1}$ as stated. In its current version this link is not clearly visible.

4. Line 102: The sentence seems to be incomplete.

5. Line 163: A reference for the uncertainty of the MWAA is required.

6. Line 912: Please check for correct reference, there is no reference Petzold et al. (2008) is the list of references.

7. In Figure 4, regression lines may be shown as full line to improve their visibility.

TYPOS

1. Line 108: It should read: "absorption Ångström exponent".

2. Line 138: Skip "with".

3. Line 165: It should read "deposited on a filter …".

4. Line 245: It should read: "the uncertainty of values …".

5. Line 253: Skip "by".

6. Line 338: It should read: "$PM_{2.5}$ fraction".

7. Line 426 – 427: I assume the Figures 4 are referenced here.

REFERENCES

Petzold, A., Veira, A., Mund, S., Esselborn, M., Kiemle, C., Weinzierl, B., Hamburger, T., Ehret, G., Lieke, K., and Kandler, K.: Mixing of mineral dust with urban pollution aerosol over Dakar (Senegal): impact on dust physico-chemical and radiative properties, Tellus, 63B, 619-634, doi: 10.1111/j.1600-0889.2011.00547.x, 2011.

---

## Author Response (AR1)

Dear editor,

We wish to thank you, the two anonymous referees, and Dr H. Moosmüller for useful comments on the manuscript. We have carefully revised the text to improve the clarity of the reading.

In particular we have made a small change to the paper title from

"Spectral- and size-resolved mass absorption efficiency of mineral dust aerosols in the shortwave: a simulation chamber study"

to

"Spectral- and size-resolved mass absorption efficiency of mineral dust aerosols in the shortwave spectrum: a simulation chamber study"

The detailed answers to the two anonymous referees are presented

Anonymous Referee #1

The paper presents needed results on dust optical properties (mass absorption efficiency and absorption Angstrom exponent) for D<2.5um and D<10um particles, relating them to chemical composition. The samples analyzed are from 12 locations in northern Africa (5 samples), Namibia (1 sample), northern China (1 sample), the Middle East (2 samples), North America (1 sample), South America (1 sample) and Australia (1 sample). I have no substantial issues with the analysis or the paper. I recommend publication after addressing the minor points below:

1) In the Abstract and somewhere earlier in the paper (before the Results section) the location where the samples were collected from should be presented. Some information about sample collection is also needed – i.e.: a) Was only one sample collected at each location, or were multiple samples collected then combined? b) Were the samples collected from locations know to be preferential sources for atmospheric dust or was the collection location just random? The latter point is important, since it's know that atmospheric dust comes from preferential locations.

To address this point, the sentence in the result section has been rewritten as "The selection of these soils and sediments was made out of 137 individual top-soil samples collected in major arid and semi-arid regions worldwide and representing the mineralogical diversity of the soil composition at the global scale. As discussed in Di Biagio et al. (2017), this large sample set was reduced by a set of 19 samples their availability in sufficient quantities for injection in the chamber. Because some of the experiments did not produce enough dust to perform good-quality optical measurements, , in this paper we present a set of twelve samples distributed worldwide but mostly in Northern and Western Africa (Libya, Algeria, Mali, Bodélé) and the Middle East (Saudi Arabia and Kuwait). Individual samples from the Gobi desert in Eastern Asia, the Namib Desert, the Strzelecki desert in Australia, the Patagonian deserts in South America, and the Sonoran Desert in Arizona have also been investigated."

The first sentence of the abstract has been changed as "This paper presents new laboratory measurements of the mass absorption efficiency (MAE) between 375 and 850 nm for twelve individual samples of mineral dust from different source areas worldwide and in two size classes".

2) Abstract, pg. 2, lines 33-34: "The size-independence of AAE suggests that, for a given size distribution, the possible variation of dust composition with size would not affect significantly the spectral behavior of shortwave absorption." Either that or the composition simply DIDN'T vary with size for this set of samples – so I'd reword this a bit.

This correction has been accepted

3) pg. 2, line 39: Need to spell out AAOD this first time that you use it.

This correction has been accepted

4) pg. 2, lines 40-41: "which is relevant to the development of remote sensing of light-absorption aerosols from space. " Not only from space! This approach has also been used extensively for AERONET (surface-based remote sensing) AAOD attribution.

The reviewer is right, the sentence has been rewritten as ", which is relevant to the development of remote sensing of light-absorption aerosols from space, and their assimilation in climate models."

5) pg. 3 line 56 contains a partial sentence ("In the solar spectrum (Boucher et al., 2013)").

This was reworded as "Albeit partially compensated by the radiative effect in the thermal infrared, the global mean radiative effect of mineral dust in the shortwave is negative both at the surface and the top of the atmosphere (TOA) and local warming of the atmosphere (Boucher et al., 2013)."

6) pg. 6, lines 162-163: "The total uncertainty, including the effects of photon counting and the deposit inhomogeneity, on the absorption coefficient measurement is estimated at 8%." Is there a basis/reference for this?

The references to the papers of Petzold et al. (2004) and Massabo' et al. (2013) have been added to the text.

7) pg. 7, line 170: It would be useful to give typical total masses and/or the fractional error/uncertainty in total aerosol gravimetric mass based on this error in filter mass.

This has been added in lines 173-175.

8) pg. 8, line 223 equation: When I read this my immediate question was: "How does this compare to the measure of total gravimetric mass?". You answer this question (appropriately) in the results section but I still think it would be useful to add a note here pointing to the fact the in your results discussion you found this agreed well with the total gravimetric mass.

The following sentence has been added "As it will be explained in the result section (paragraph 3.1), the values of MCdust estimated from Equation 4 were found in excellent agreement with the measured gravimetric mass on the filters"

9) pg. 10: Do you have any estimate / sense of how well the dust suspended by this "shaking" compares to the dust lofted by winds?

Prior to any scientific analysis, we have dedicated a lot of energy to investigate the realism of our dust generation system, both in terms of the composition and the size distribution of the dust aerosols. These results are reported in two papers:

- Di Biagio, C., P. Formenti, S. A. Styler, E. Pangui, and J.-F. Doussin (2014), Laboratory chamber measurements of the longwave extinction spectra and complex refractive indices of African and Asian mineral dusts, Geophys. Res. Lett., 41, doi:10.1002/2014GL060213. – Figure 2 and discussion

But in particular in

- Di Biagio, C., Formenti, P., Balkanski, Y., Caponi, L., Cazaunau, M., Pangui, E., Journet, E., Nowak, S., Caquineau, S., Andreae, M. O., Kandler, K., Saeed, T., Piketh, S., Seibert, D., Williams, E., and Doussin, J.-F.: Global scale variability of the mineral dust long-wave refractive index: a new dataset of in situ measurements for climate modeling and remote sensing, Atmos. Chem. Phys., 17, 1901-1929, doi:10.5194/acp-17-1901-2017, 2017

where we have dedicated two paragraphs (5.1. Atmospheric representativity: mineralogical composition and 5.2 Atmospheric representativity: size distribution) to show how the composition and the size distribution of the generated dust are well representative of those of real dust in the atmosphere, which makes the laboratory experiments well suited for studying the dust optical properties.

In order to stress this point further, without repeating results already presented in these two publications, we have added the following sentence in paragraph 3 "Di Biagio et al. (2014; 2017) have demonstrated the realism of the generation system concerning the composition and the size distribution of the generated dust with respect to the properties of mineral dust in the atmosphere".

10) pg. 11, line 209: Mauritania is not listed as one of the sample site locations in Table 3. ?

For Mauritania, we only have chemical composition but not optical measurement results. The reference to this sample has been taken out of the paper.

11) pg. 11, lines 309-310 (an onward): The results here are said to agree well with that found for atmospheric aerosols in other studies, but the values in these studies is not given so this feels very hand-waving and unconvincing. Are you referring to the values given in Table 5? If so, please refer directly to them. If not, the comparison here needs to be more quantitative (discuss numbers from the literature vs. what is found here).

Additional text and values have been added in lines 321-392 to address this point

12) pg. 13, lines 353-354: If you have results for Niger why not show them?

We do not have optical results for Niger

13) pg. 14, lines 387-388: MAE doesn't vary linearly inversely with wavelength, it varies linearly inversely with the log of the wavelength (hence our ability to use the AAE relationship).

The reviewer is correct, this has been corrected as "and decrease in a linear way with the logarithm of the wavelength"

14) pg. 15, lines 429-434: A few things: a) "satisfactory" and "loose" are not quantitative terms, nor are they really appropriate for a scientific paper. What constitutes "satisfactory"? Best is to just give the correlations. b) The high correlation coefficients for PM2.5 are really driven by one high data point and so are probably not very robust.

These sentences have been reworded as "Regardless of the size fraction, the correlation between the MAE values and the percent mass of total elemental iron are higher at 375, 407 and 532 nm" and "At these wavelengths, linear correlations with the mass fraction of iron oxides are low in the $PM_{10.6}$ mass fraction ($R^2$ up to 0.38-0.62), but higher in the $PM_{2.5}$ fraction ($R^2$ up to 0.83-0.99)"

15) pg. 16, lines 455-456: How can fine-mode-only AAOD be GREATER than total aerosol AAOD?

Total and fine were inverted by mistake, this is now corrected

16) pg. 17, lines 487-488: As written this implies Solomon et al. varied SSA by 5%.

For, say, SSA of 0.9, that they varied SSA by 0.045. That is, the co-albedo (or absorption) was varied by 45% (0.1+/-0.045). This is the proper comparison to make to the variation in MAE that you calculate

The reviewer is right. The sentence has been corrected as "As a comparison, Solmon et al. (2008) showed that varying the single scattering albedo of mineral dust over western Africa by ± 5%, that is, varying the co-albedo (or absorption) by 45% (0.1± 0.045) could drastically change the climate response in the region."

7) Overall: Some editing is needed for language throughout. Here I list some that stood out to me – all small stuff but editing would help readability:

pg. 3, line 67-68: "in the last ten years or so" (too casual for scientific writing)

removed pg. 3, line 74: "A significant body of observations have been performed."

Replaced by "A significant number of observations have quantified"

pg. 8, lines 214-215: "The linear deconvolution, performed the Athena IFEFFIT free-ware analysis program (Ravel and Newville, 2005), provided with the proportionality factors alpha_i representing the mass fraction of elemental iron to be assigned to the i-th standard mineral." (I found this sentence nearly impossible to follow)

Replaced by "The linear deconvolution has been performed with the Athena IFEFFIT freeware analysis program (Ravel and Newville, 2005). This provided with the proportionality factors    i representing the mass fraction of elemental iron to be assigned to the i-th standard mineral."

pg. 10, line 253: "the chamber was evacuated by to" (delete "by")

Corrected pg. 10, line 259: "dust particles produced was" –> "dust particles produced were"

The sentence was corrected as "The dust particles produced by the mechanical shaking, mimicking the saltation processing that soils experience when eroded by strong winds, were injected in the chamber by flushing the flask with N2 at 10 L min-1 for about 10-15 min, whilst continuing shaking the soil."

pg. 10, lines 279 "dust on filter for" –> "dust on the filter for"

Corrected as "dust on the filter membranes for subsequent chemical analysis"

pg. 10, line 280: "by placing the loaded filter holders": This reads as if you are placing LOADED FILTERS (vs holders with blank filters in them, which is what I assume you mean). Reword.

Corrected as "by placing the filter holders loaded with filter membranes"

pg. 11, line 206: "the origin of used dust samples". I think this should be "the origin of our dust samples", yes?

Corrected as "the origin of dust samples"

pg. 12, line 315: "Henceforth, and contrary to the soil samples"

pg. 12, lines 317-318: "could reflect more that of the parent sedimentary soil than not the other samples."

These two sentences were rewritten as "This powder is likely to be injected in the chamber with little or no size fractionation. Henceforth, the aerosol generated from it should have a closer composition to the original powder than the other samples.

pg. 12, line 335: "An the exception"

Corrected as "Exceptions are"

pg. 13, lines 353-354: "As a matter of fact, the number fraction of particles in the size classes above 0.5 µm in diameter are different in the dust aerosol generated in the Alfaro et al. (2004) study with respect to ours."

Corrected as "As a matter of fact, the number fraction of particles in the size classes above 0.5 µm in diameter is different in the dust aerosol generated in the Alfaro et al. (2004) study with respect to ours."

pg. 13, line 364: "On the contrary" –> "In contrast

Corrected pg. 13, line 365: "These differences could yield either to difference in the"

Corrected as "These differences could either be due to difference in the chemical composition and/or in the total mass in the denominator of Equation 8."

pg. 15, lines 415-417: "using a power-law function fit as from Equation 2, provides withthe values of"

Corrected as "using the power-law function fit (Equation 2)"

pg. 16, line 448: "The size-dependence, yielding significantly higher MAE values"

Corrected as "The size-dependence, yielding significantly higher MAE values in the fine fraction (PM2.5) than in the bulk (PM10.6) aerosol,"

pg. 17, line 470: "A closer look to observations" –> "A closer look at observations"

Corrected as "These differences could either be due to difference in the chemical composition and/or in the total mass in the denominator of Equation 8."

pg. 17, line 478: "our estimated MAE average at" –> "our average MAE values are"

Corrected pg. 17, line 490: "As Moosmuller et al." –> "As in Moosmuller et al."

Corrected pg. 18, line 503-504: "pointing out to the" –> "pointing out the

Corrected

Anonymous Referee #2

GENERAL COMMENT

The manuscript presents important results from a carefully designed and conducted study on the light-absorbing properties of mineral dust from various origins. Dust samples collected at the different source regions have been re-suspended in an aerosol chamber and characterized with respect to microphysical, optical and chemical properties by state of the art methods. The study provides urgently needed knowledge on the multi-spectral light absorbing properties of mineral dust and for sure deserves publication in ACP. The manuscript is well structured, the methods are described in necessary detail and the referenced literature reflects the current state of knowledge. I recommend publication after the following minor revisions have been considered.

SPECIFIC REMARKS

1. In the experimental protocol section, the potential impact of gravitational settling on the re-suspended fraction of the dust samples is mentioned.

Given the instrumentation list, the size distributions of the airborne dust samples were monitored during the runs of the experiments. It appears obvious to control the change of the size distribution during the experiment time in the chamber. Since the mass concentration in the chamber decreased very rapidly after injection, whereas the chemical composition of the dust samples was determined from bulk samples, it would be important to know if the airborne fraction sampled for the determination of optical properties features the same chemical properties as the bulk samples. At least a discussion of this potential source of uncertainties should be presented, along with a plot showing the change of the size distributions during the experiment time. The current analysis starts from the assumption that the dust bulk properties represent also the properties of the sampled airborne fractions. However, is this really justified?

The reviewer is right when saying that the samples collected for the investigation of the chemical composition are time-integrated and henceforth might reproduce dust with varying size distributions. Examples of the time variability of the size distributions are provided by Di Biagio et al. (2017) - Figures 7 and 5S in the supplementary material. These figures show that, after the very strong initial depletion of particles larger than 10 μm in diameter (when no sampling on filters was performed), the number concentration decreases at a rate, which is almost independent of size, suggesting that no significant distortion of the particle size distribution occurs after the most significant removal at the beginning of the experiment.

We also would like to stress that our generation system allows to generate a dust aerosol from a soil, and that this dust aerosol is injected in the chamber, not the soil. Henceforth, when talking about "bulk" we refer to the total aerosol fraction, sampled from the chamber without any size segregation other than that imposed by the cutoff the sampling lines. The fine fraction corresponds to the same dust aerosols, but sampled by an impactor with a 2.5 diameter cutoff.

2. In section 3.2, the variability of dust optical properties with particle size is discussed. The authors found no statistically significant size-dependence of the absorption Ångström exponent (AAE), whereas the absolute values of the mass absorption efficiencies (MAE) show large differeces between the PM2.5 and PM10 fractions with larger values for the fine mode fraction.

These findings imply that the relative chemical composition with respect to light-absorbing compounds does not change between the size fractions (similar AAE values), whereas the differences between the MAC values indicate that coarse mode particles contain more non-absorbing matter than fine mode particles (higher MAE values for smaller particles). This however, this is in contrast to the assumption that the chemical composition is uniformly distributed across the particle size distribution. Here, a detailed discussion is requested.

The reviewer's statement is in error as for a given mineral composition (given effective complex refractive index) the MAE depends strongly on size, decreasing with size at larger sizes. So having a smaller MAE for PM10 than for PM2.5, does not necessarily imply that PM10 contains less absorbing matter than PM2.5, but may just be due to the change in size distribution. This is shown by our calculations. An independent example of the size dependence is also given by Fig. 1 of Moosmuller et al. (2009).

Moosmüller, H., R. K. Chakrabarty, and W. P. Arnott (2009), Aerosol light absorption and its measurement: A review, Journal of Quantitative Spectroscopy and Radiative Transfer, 110(11), 844-878.

3. In section 4, it is discussed that the potential impact of light absorption by mineral dust may play an important role even after long range transport. Cited studies all refer to observations in China.

However, there is another detailed study on this effect available for the pollution plume of Dakar mixing with mineral dust which also includes the variation of the AAE during mixing (Petzold et al., 2011). The authors may consider including this study.

This reference has been added to the manuscript.

MINOR COMMENTS

1. The list of references contains various references which are not cited in the manuscript. This should be checked, I found the following but there may be more: Anderson et al., 1998, Andrews et al., 2006, Arnott et al., 2005, Collaud Coen et al., 2010, Petzold et al., 2013.

Corrected

2. Line 56: The sentence seems to be incomplete.

The sentence was corrected as "Albeit partially compensated by the radiative effect in the thermal infrared, the global mean radiative effect of mineral dust in the shortwave is negative both at the surface and the top of the atmosphere (TOA) and local warming of the atmosphere (Boucher et al., 2013)."

3. Line 97 – 98: The basic unit of mass concentrations is gm-3. Using this unit, then the unit of the combined property MAE is m2 g-1 as stated. In its current version this link is not clearly visible.

4. Line 102: The sentence seems to be incomplete.

The sentence was removed

5. Line 163: A reference for the uncertainty of the MWAA is required.

This has been added

6. Line 912: Please check for correct reference, there is no reference Petzold et al. (2008) in the list of references.

Corrected

7. In Figure 4, regression lines may be shown as full line to improve their visibility.

Done

TYPOS

1. Line 108: It should read: "absorption Ångström exponent".

Corrected

2. Line 138: Skip "with".

Corrected

3. Line 165: It should read "deposited on a filter ...".

Corrected

4. Line 245: It should read: "the uncertainty of values ...".

Corrected

5. Line 253: Skip "by".

Corrected

6. Line 338: It should read: "PM2.5 fraction".

Corrected

7. Line 426 –427: I assume the Figures 4 are referenced here.

Corrected

REFERENCES

[revised manuscript text omitted]